# Current Trends of Bacterial and Fungal Optoproteins for Novel Optical Applications

**DOI:** 10.3390/ijms241914741

**Published:** 2023-09-29

**Authors:** Carolina Ramírez Martínez, Leonardo S. Gómez-Pérez, Alberto Ordaz, Ana Laura Torres-Huerta, Aurora Antonio-Perez

**Affiliations:** Escuela de Ingeniería y Ciencias, Tecnológico de Monterrey, Campus Estado de México, Av. Lago de Guadalupe KM 3.5, Margarita Maza de Juárez, Ciudad López Mateos, Atizapán de Zaragoza 52926, Estado de México, Mexico; a01272199@tec.mx (C.R.M.); a01373058@tec.mx (L.S.G.-P.); alberto.ordaz@tec.mx (A.O.); atorresh@tec.mx (A.L.T.-H.)

**Keywords:** optogenetics, photoreceptors, bioluminescence, photoswitchable proteins, optocontrol

## Abstract

Photoproteins, luminescent proteins or optoproteins are a kind of light-response protein responsible for the conversion of light into biochemical energy that is used by some bacteria or fungi to regulate specific biological processes. Within these specific proteins, there are groups such as the photoreceptors that respond to a given light wavelength and generate reactions susceptible to being used for the development of high-novel applications, such as the optocontrol of metabolic pathways. Photoswitchable proteins play important roles during the development of new materials due to their capacity to change their conformational structure by providing/eliminating a specific light stimulus. Additionally, there are bioluminescent proteins that produce light during a heatless chemical reaction and are useful to be employed as biomarkers in several fields such as imaging, cell biology, disease tracking and pollutant detection. The classification of these optoproteins from bacteria and fungi as photoreceptors or photoresponse elements according to the excitation-emission spectrum (UV-Vis-IR), as well as their potential use in novel applications, is addressed in this article by providing a structured scheme for this broad area of knowledge.

## 1. Introduction

Cells continuously process information from their environment and yield complex responses. Furthermore, if these stimuli are not correctly processed through specific and well-timed molecular mechanisms, cell survival is put at risk. Light plays a decisive role in most forms of life, as it stimulates various essential processes in organisms [1]. Phototaxis, photomorphogenesis, conversion of light energy into biochemical energy, light signaling cascades, phototropism and photocontrol are some examples of light-mediated biological processes. Photoproteins are light-reversible, activated molecules that affect cellular processes upon light stimulation [2]. These kinds of proteins can absorb photons emitting in the wide range between 300 nm (ultraviolet, UV) and 800 nm (near-infrared light, NIR) to trigger photochemical reactions. There are two main mechanisms described for photon absorption, either intrinsic tryptophan antennas or chromophores bound to photoreceptors [3]. Sensory photoreceptors translate photon absorption by their chromophore into changes in their biological activity, for instance, enzymatic activity or interaction with other biomacromolecules [4].

It was considered that bacteria were not organisms that responded to light stimuli, except for phototrophs that use sunlight as an energy source [5]. Numerous sequences corresponding to photoreceptors among nonphototrophic species have currently been identified [4]. Many efforts have been made to elucidate the control mechanisms exerted by light on various biological processes in bacteria. For example, it has been observed that some pathogenic bacteria develop their propagation in a particular location with the aim of increasing their invasive capacity in a new host through the sensing of light that they perceive through photoreceptors [6,7]. On the other hand, it has also been verified that the degree of light to which bacterial communities are exposed is a determining factor for the type of growth that bacterial communities will develop, whether in a single-cell motile state or multicellular state, which is characteristic of the structure of natural biofilms [8].

In fungi, light is an important regulatory cue; it is involved in the regulatory control of carotenogenesis, sporulation, circadian rhythmicity, morphogenetic pathways, the synthesis of hydrolytic enzymes, and the production of primary and secondary metabolites, such as penicillin [9,10,11,12,13,14]. Most fungi contain several photosensory systems that respond to blue light and other light intensities and wavelengths. Therefore, they can even distinguish between colors owing to their versatile photoreceptors, such as the white-collar proteins and cryptochromes for blue light, opsins for green light and phytochromes for red light [15,16,17]. Additionally, in genome-wide transcription studies, it has been demonstrated that large groups of genes are controlled by light [18].

Some proteins of bacterial or fungal origin have shown great potential in various applications due to their wider spectrum of functions, as occurs when comparing microbial opsins with respect to others of animal origin. This article was conducted by integrating the latest advancements in bacterial and fungal optoproteins for novel optical applications. The result was based on comprehensive research using the most significant articles in the field and highlighting the most exciting discoveries made in various scientific disciplines. We first recapitulate fundamental aspects of photoproteins, their classification focused on bacteria and fungi, and their applications. We will address relevant applications in the areas of biosensors, control of bioprocesses, materials, genome editing and imaging, among other interesting uses.

## 2. Photoreceptors Classification

Photoproteins can be divided into approximately ten distinct families according to the chemical basis of the chromophores (natural or synthetic origin) used for light absorption, the structure, and the signaling mechanisms of sensory photoreceptors. Photoreceptor types using different chromophores have evolved for different purposes and are receptive to different wavelengths of light (Figure 1). The categories are tryptophan antennas, flavin adenine dinucleotide (FAD), p-coumaric acid, retinal or vitamin A derivatives, the Cys-Tyr-Gly triad (DronpaN), cobalamin or vitamin B12, and bilin analogs. 

Light-oxygen-voltage (LOV) proteins form a group of photoreceptors that bind different flavins such as flavin mononucleotide (FMN) or flavin adenine dinucleotide (FAD) and respond to blue illumination (400–473 nm) eliciting several physiological responses across archaea, bacteria, protists, fungi, and plants. LOV receptors also serve as genetically encoded actuators in optogenetics for the spatiotemporally precise control of protein expression, cellular state, and processes [19]. The light responses of LOV are enhanced due to photochemical and structural mechanisms; they have been characterized in detail in [20,21]. LOV domains form internal protein-flavin adducts that generate conformational changes that control effector function [22]. In bacteria, LOV proteins are intrinsically related to a range of signal transduction output domains, such as histidine kinases, diguanylate cyclase-phosphodiesterases, and DNA-binding domains [23].Plant phytochromes (Phys) constitute a large receptor family sensitive to red and far-red light that occur in plants and employ as chromophores linear tetrapyrroles (bilins). These proteins have a photointerconvertible switch between two forms throughout the photoisomerization of the chromophores. These two forms can absorb red or far-red light and are designated as Pr and Pfr, respectively. They act as proximity sensors to modify plant growth and development [24]. Phytochromes are structurally classified according to the number of domains in their photosensory core module (PCM). Members of the “canonical” phytochrome subfamily (from bacteria, cyanobacteria, and plants) contain three domains in their PCM (PAS, Per-ARNT-Sim; GAF, cGMP phosphodiesterase-adenylate cyclase-FhlA; PHY, phytochrome-specific domain) and an output effector module, which is typically represented by histidine kinase (HisK). Despite the amino acid sequences of these domains showing low sequence similarity, their structures share a common topology.Algal phytochromes cover the entire visible and near-infrared spectrum. Compared to land plants, algae have a different type of phytochrome, since red and far-red light do not penetrate water to depths greater than a few meters. This kind of protein exhibits a great diversity of photocycles (yellow/ far-red, orange/far-red, red/far-red, blue/ far-red, red/blue, far-red/green, etc.) that depend on the type of algae [25]. Similar to plant phytochromes, their photosensory core module (PCM) is canonically constituted of the conserved consecutive PAS, GAF, and PHY domains. However, the output modules in the C-terminal are more diverse in their composition, with histidine kinases or related domains as well as the presence of one or more response regulator receiver (REC) domains [26]. The proteins in this family are CparGPS1, EsilPHL1, GwitGPS1, NpyrPHY1, PcolPHY1, DtenPHY1, and TastPHY1. Knowledge of the details of the functions of this type of protein will be crucial for the development of applications.Bacteriophytochromes (BphP) were described for the first time in the nonphotosynthetic bacterium *Deinoccocus radiodurans,* and over fifty have been found in purple bacteria. These proteins play important roles in intracellular signaling, regulating the expression of respiration and photosynthetic complexes. Most phytochromes are dimeric proteins, and all exhibit a modular domain of architecture PAS–GAF–PHY. This photosensory core is conserved between phytochromes in plants, bacteria, and fungi [27]. They are characterized by interacting with biliverdin IXα as an endogenous cofactor; the bilin chromophore is attached to the PAS or GAF domain. Different output domains are found on the C-terminal side of the photosensory core. In cyanobacteria and bacteria, the output module commonly consists of histidine kinase domains (PHY–histidine kinase modules), and this is the first part of a two-component signaling mechanism [28]. They have a relatively simple but diverse modular architecture that can adopt two spectroscopically different metastable states depending on the light conditions [29]. The states generated during the photocycle are generically called Pr (red light-absorbing) and Pfr (far-red light-absorbing) states. As can be seen, this denomination specifies the wavelength that stimulates the conformational changes in each bacteriophytochrome, and these wavelengths can vary between phytochrome species [30]. The complexity of the Pr→Pfr photoconversion process increases from bacteria to cyanobacteria to plants [27].Cyanobacterial phytochromes (Cphs) have been recognized as the most structurally and functionally diverse subgroup of the phytochrome photoreceptor superfamily. Cphs contain multiple photosensory modules that function together to fine-tune light responses [31]. The most complex type of a prokaryotic member of this family possesses a generalized photosensory module integrated by PAS–GAF–PHY domains as bacteriophytochromes [32]. However, the overall architecture of cyanobacterial phytochromes varies from the typical PAS-GAF-PHY, and some lack the upstream PAS domain while maintaining the PHY domain [33]. Cyanobacteria phytochromes use phycocyanobilin (PCB) or phytochromobilin (PΦB).Cyanobacteriochromes (CBCRs) are linear tetrapyrrole-binding photoreceptors with a high diversity in the colors of light they sense. Most phytochromes require an architecture composed of three domains: GAF domain (cGMP-phos- phodiesterase/adenylate cyclase/FhlA), PAS domain (Per/ARNT/Sim), and PHY domain (phytochrome-specific). CBCRs lack the PAS and PHY domains, and only the GAF domain is needed for chromophore incorporation and proper photoconversion of the CBCRs [34]. The GAF domain can attach four kinds of linear tetrapyrrole chromophores, biliverdin (BV), phytochromobilin (PFB), phycocyanobilin (PCB), and phycoviolobilin (PVB), which leads to four distinctive color-tuning mechanisms [35]. CBCRs are more spectrally diverse than phytochromes, CBCRs covering the entire UV-to-visible spectrum and near-infrared wavelengths; they function as light sensors in the 300–750 nm region [26,35].Xanthopsin are also called photoactive yellow proteins (PYPs), and they are characterized by the binding of the chromophore from cinnamic acid thioester- to coumarin-based ligands [36,37]. PYP is a model bacterial photoreceptor that has an evolutionary domain of the PAS superfamily [35] (Per-Arnt-Sim, named after the Per, Arnt, and Sim proteins, in which they were first observed) and is restricted to a few bacterial phyla and distinct from other PAS domains. These proteins play the role of photoreceptors for signal transduction in gene expression, motility, and biofilm formation [38]. A light-absorbing chromophore in the protein absorbs blue light and changes its shape from a nearly straight conformation (trans) to a bent conformation (cis) in less than a picosecond. Then, the activation of other sensing proteins in the bacterium occurs, which controls the direction of the bacterium. Finally, the chromophore shifts back to the straight conformation in less than a second, ready to sense another blue photon [39]. PYP-like proteins were identified first in halophilic purple bacteria, and through phylogenetic studies, they can be found in orders such as Myxococcota, Proteobacteria, and some other more distant bacterial phyla [38].Advantages of this type of protein include its small size, relatively high brightness, and photostability of synthetic fluorescent probes, such as the PYP from the halophilic phototrophic bacterium *Halorhodospira halophila* [37]. Proteins fused to tags on PYPs can be quantified due to the acquisition of a yellow color upon adding a precursor of the chromophore. The production of the target protein can be conducted via visual inspection within a few seconds, as well as with the aid of a spectrometer within a few minutes [40]. Other applications of PYPs include their employment to image proteins in a cellular environment as tags and fluorescent probes [41], films for optical switching on circuits [42], and optogenetic modules exerting allosteric regulation in a light-dependent manner [43], among others.Cryptochromes are photolyase-like flavoproteins that mediate blue-light regulation of gene expression and photomorphogenic responses. These receptors were originally discovered in *Arabidopsis thaliana*, but were later found in other plants, microbes, and animals. Despite cryptochromes being structurally related to photolyases since both are flavoproteins, only photolyases catalyze light-dependent DNA repair. On the other hand, cryptochromes have carboxyl-terminal extensions of variable length, and usually they have lost or reduced DNA repair activity and play a role in signaling. However, there are several examples of photolyases, including some from fungi, that have a dual function as a DNA-repair enzyme and photoreceptor. Genes encoding bacterial and fungal cryptochromes are found in different taxonomic groups. Very few cryptochromes from bacteria have been characterized [44]. The FeS-CPD (iron-sulfur bacterial-cryptochrome/photolyases) includes phrB from *Agrobacterium*. While members of the cluster CryPro (proteobacterial cryptochromes) are highly abundant in proteobacteria and cyanobacteria [45]. On the other hand, Cry-DASH proteins are capable of binding single-stranded and double-stranded DNA or RNA but do not always provide photorepair activity. Cry-DASH proteins participate in the regulation of development and pigment accumulation and the regulation of the circadian clock through a CRY-dependent oscillator [46]. CRY-DASH proteins have been found in photosynthetic cyanobacteria and also in nonphotosynthetic bacteria, fungi, plants, and animals. The shape of cryptochromes changes in response to blue-light perception from an inactive monomeric state to an active homodimeric state.UV-B receptors. The first protein identified of this kind is UVR8 (UV Resistance Locus 8) from *Arabidopsis thaliana*. For this type of protein, no homologues have been identified in bacteria or fungi; as was observed, a very low percentage of identity was identified in phylogenetic studies. The UVR8 family proteins have a highly conserved domain called VP, which is useful for the interaction of the COP1 protein, a key event in UV-B signaling. Additionally, there is a conserved peptide repeat called GWRHT (Gly-Trp-Arg-His-Thr), that generates a triad of closely packed tryptophans for UV-B photoreception [47]. Unlike other photoreceptors, UVR8 does not contain an external chromophore, but it uses tryptophan side chains to sense UV-B light. The homodimer of UVR8 dissociates into two monomers in response to UV-B irradiation, and the reaction is reversible by removing the UV-B stimulus [48]. Each monomer interacts with signaling partners to regulate the expression of genes that trigger UV protective mechanisms [49]. This behavior is useful for the design of UV-B-based optogenetic tools. Since it does not require external chromophores, it can be used in models other than plants. Optogenetic tools have been used in mammalian cells to regulate gene expression through the assembly of a chimeric transcription factor, and to localize secreted proteins of interest [50]. Furthermore, in yeast, there was a report on the expression of an activation system based on the organization of the genome [51].BLUF (blue-light sensors using flavin-adenine dinucleotide) proteins are flavin-nucleotide-binding cryptochromes that control enzyme activity or gene expression in response to blue light [52]. The structural element responsible for the detection of light in these photoreceptors is the BLUF domain (15 kDa); it transmits the light-induced signal to downstream protein modules via intermolecular or intramolecular interactions. Unlike other photoreception mechanisms, such as those developed by rhodopsin or phytochromes, the photoactivation of the BLUF proteins does not involve major structural changes in the chromophore FAD [53]. The interaction with the FAD chromophore is mainly mediated by nonovalent contacts between the chromophore and various amino acids of the BLUF domain; however, all BLUF proteins show conserved Tyr, Gln, and Met residues intimately related to function [52]. In particular, the signaling status of BLUF proteins arises from the structural change in the hydrogen bonding network between FAD, Gln, and Tyr, conserving the active site [54].Opsins are transmembrane proteins that use the retina to respond to different light bands in the UV-to-red spectral region. These proteins have different functions, such as the control of proton gradients, the maintenance of membrane potential, ionic homeostasis, and the modulation of flagellar motor rotation. Opsins can be found across all kingdoms of life, and they require the incorporation of a vitamin A-related organic photon-absorbing cofactor to enable light sensitivity, a complex referred to as rhodopsin [55]. Opsin genes are divided into two families: microbial opsins (type I) and animal opsins (type II). Even though no sequence similarity is shared between microbial and animal rhodopsins, they share a common architecture of seven transmembrane α-helices (TM) with the N- and C-terminus facing out- and inside the cell, respectively. Most microbial rhodopsins function as light-sensing receptors and light-driven ion pumps for the active transport of ions (H^+^, Na^+^, Cl^−^, etc.). On the other hand, some microbial rhodopsins function as light-gated ion channels, also called channelrhodopsins, mainly used as optogenetic tools in neuroscience.

Additionally, another group called enzyme rhodopsins, which are light-activated, comprises the histidine kinase rhodopsins (HKRs) and rhodopsin guanylyl cyclases (Rh-GC).

## 3. Bioluminescent Proteins

There are additional types of processes other than those previously discussed, specifically for the “emission or generation” of light. Bioluminescence is the term used to describe the visible spectrum of light that living organisms emit, a characteristic that has been evolutionarily conserved primarily in marine organisms, insects, and some species of bacteria and fungi [56]. When an enzyme called luciferase catalyzes the oxidation of the tiny molecule luciferin to create an excited-state species that emits light, bioluminescence is produced. The study of bioluminescent reactions across numerous species has uncovered an astounding number of mechanisms that drive light emission, as well as the substrates, cofactors, and enzymes that catalyze these reactions. As a result, rather than referring to structural or chemical principles, the labels “luciferase” and “luciferin” were used to describe the enzymes and substrates of bioluminescent processes [57]. During biochemical oxidation, oxidized “luciferins” transition to an excited state and emit light when they return to the ground state. These biochemical reactions may be induced by the organism itself, known as autogenous bioluminescence. There are organisms that can house and control luminous bacterial colonies in specialized light organs, this phenomenon is called bacteriogenic bioluminescence [56].

Although there are more than 30 known bioluminescent systems, only 11 have been well characterized [58], including the newly solved fungal [59] and bacterial systems [60]. The bioluminescent proteins have not yet been fully identified for the majority of bioluminescence systems. A common way to classify bioluminescence systems is through the type of compound that acts as luciferin, as well as the reaction mechanism that is activated and develops from contact with this compound. D-luciferin-dependent systems have commonly been found in insects; systems activated by tetrapyrrole-based luciferins have been identified in dinoflagellates and euphausiids; and coelenterazine-dependent systems have been observed in numerous independent and phylogenetically distant groups of marine organisms. Cypridina luciferin-dependent systems were identified in the bioluminescent midshipman fish Porichthys and the ostracod Cypridina [61].

The majority of known bioluminescent bacteria are gram-negative, facultatively anaerobic, and symbiotic [62]. All bioluminescent bacteria use the same fundamental process for light emission, in which photons are generated in an array of reactions requiring oxygen, nicotinamide adenine dinucleotide (NADH), FMN, and myristic aldehyde. Bacterial luciferase is a heterodimeric enzyme that catalyzes the oxidation of reduced flavin mononucleotide (FMNH−), the mono-oxygenation of a long-chain aliphatic aldehyde, and the reduction in molecular oxygen, resulting in the formation of a hydroxide of an excited state, FMN-4a-hydroxide, which is the luciferin in the bacterial bioluminescent reaction. As this luciferin relaxes to the ground state, the free energy is released as light with an emission maximum at 490 nm. After the release of one water molecule, the catalytic cycle is completed, and FMN returns to its oxidized ground state [63]. Bacterial luciferases are encoded in the lux operon together with the enzymes that catalyze the manufacture of luciferin, and they are composed of two polypeptide chains that combine to create a 75 kDa protein complex [64]. The classic lux operons adopt the CDAB(F)E(G) gene order, and luxA, B, C, D, E, and G are regarded as the main genes, despite the fact that the lux operon organization demonstrates a greater degree of variation [65].

There are approximately 81 species of fungi known to be bioluminescent [66]. According to the information acquired about the bioluminescent mechanisms in fungi, there is only one bioluminescent mechanism in fungi, and all luminescent fungi share the same kind of luciferin and luciferase [67]. The four enzymes involved in the fungal bioluminescence mechanism were discovered in *Neonothopanus nambi*. The enzyme, Hispidin synthase, (*hispS*) converts caffeic acid to hispidin, then hispidin-3-hydroxylase (*h3h*) hydroxylates hispidin to provide 3-hydroxy hispidin (fungal “luciferin” compound), and fungal luciferase (*luz*) oxidizes 3-hydroxy hispidin to receive an unstable high-energy intermediate that emits green (~530 nm). The final product can be recycled into caffeic acid by caffeylpyruvate hydrolase (*cph*) [59]. It has been discovered that fungi have distinct regulating systems for bioluminescence; however, its mechanistic details have not been completely elucidated. For example, a biological circadian rhythm was revealed by the profile of bioluminescent elements in mycelium; this suggests a fungus strategy for attracting insects to spread spores with lower energy consumption [68]. In another study, the authors resolved the phylogenetic relationship between major groups of bioluminescent fungi and constructed an evolutionary model that pinpointed changes in the luciferase cluster across all bioluminescent fungi, and additional genes involved in bioluminescence were suggested. The variety of luciferase sequences found in these fungi may supplement ongoing studies on luciferase engineering and optimization [69].

Numerous efforts have been made for decades to understand in detail the molecular mechanism of bioluminescence in bacteria and fungi. Although a consensus has been reached on the basic elements and processes, omics and structural studies are still ongoing and will continue to be developed to better understand these systems under the microenvironmental conditions of regulation. Along with these studies, mechanistic knowledge will increase at the promoter, gene, and protein structural levels, which will surely lead to more new applications.

## 4. Photoswitchable Proteins

Fluorescent properties in proteins were first discovered in the GFP from *Aequorea victoria* by Shimomura, Johnson, and Saiga in 1962. Later, the X-ray structure of this protein was solved by two groups, Ormö et al., 1996 [70] and Yang et al., 1996 [71], establishing the structural basis of fluorescent proteins. Along with the structural information of the 11-strand ***β***-barrel structure and the presence of a 4-(p-hydroxybenzylidene)-5-imidazolinone (p-HBI) chromophore, phenomena such as photoactivation and photoconversion were reported, observing that the absorption-emission spectra of the protein were modulated in an irreversible way or even through a complete deactivation caused by covalent modifications.

Similarly, photoswitching was described as the process where light induces conformational changes in the chromophore but in a reversible manner, allowing the shifting between the OFF and ON states. In this regard, photoswitchable proteins are able to modulate their capability of interacting with light at specific wavelengths by isomerization of the chromophore. Despite the nature of the chromophore and its interactions with light that drive the photoswitching process, several studies [72,73,74] have pointed out the relevance of the protein structure, the chemical environment (pocket) where the chromophore is located, the cis/trans conformations, photoisomerization mechanisms, photochromism, and protein architect modulations that undergo as a response to the light changes and photoswitching. Generally, the cis isomer is considered the ON state, while the trans isomer is referred to as the OFF state; nonetheless, the stabilization of the isomers by specific residues from the protein could lead to inverted behaviors where the OFF states are associated with the cis isomers and vice versa. Therefore, the implications of the chemical environment favored by the protein enclosure are relevant in the photoswitching, allowing for the stabilization of the chromophore in stable and metastable states.

Two mechanisms are relevant in the discussions of photoisomerization of chromophores: the One Flip Bond (OFB) and the Hula Twist (HT) (Figure 2A). OFB is considered to be a volume-demanding mechanism since the pocket where the chromophore residues are located must be large enough to allow the isomerization of the double bond. On the other hand, HT requires a lower volume due to the simultaneous rotation of sigma and pi bonds being favored under less steric hindrance conditions [75]. Since the mechanisms are different in the number of rotational bonds and the direction of the rotation, the isomers obtained in one way or another are also different. Therefore, establishing the correct photoisomerization mechanism is relevant to understanding the outcome of the photoswitching process. 

Most photoswitchable proteins reported in the literature come from engineered versions of GFP, such as dronpa, padron, rsCherry, IrisFPs, and Dreiklang [72]; some of them even have additional features, such as the inverse behavior of photoswitching (the cis isomer being the OFF state), irreversible photoconversion from green to red, and decoupled spectra of excitation and photoswitching. In spite of the wide range of GFP-based proteins and their properties, few other photoswitchable proteins have been described with an unusual structure. In fact, photoswitching has been generally associated with the fluorescence effect of proteins, emphasizing the relationship with GFP-like proteins and excluding other proteins with such a property.

According to the Opto database, photoswitchable proteins from bacteria and fungi are allocated in six categories (Table 1): cyanobacteriochromophore, BLUF, LOV, fluorescent, cobalamin-binding, and phytochrome. The data updated to 2023 show that LOV and phytochrome domains are the more abundant groups, with phytochromes constituting only examples of bacteria; nonetheless, examples of fungi phytochromes have been reported since 2005 [31,76]. Despite the lack of opsin-based switches and fungi phytochromes, the information acquired from the database represents the diversity of photoswitchable proteins present in the bacteria and fungi kingdoms. Similarly, the bacteriophytochromes, LOV, and BLUF, present in fungi and bacteria, are the most enriched photosensory modules studied, pointing out the relevance of these three families as photoswitchable proteins in the previously mentioned kingdoms.

### 4.1. Photoswitching Mechanism of Bacteriophytochromes

The first BphP discovered was obtained from the cyanobacterium Fremyella diplosiphon and named RcaE [78]. Unlike plant phytochromes, which bind the chromophore to a cysteine in the GAF domain, in RcaE the chromophore is attached to the protein structure through a buried cysteine present in the PAS domain [30,79,80]. This characteristic has been generally observed among BphP from fungi and bacteria (Figure 2B), and regardless of the nature of the phytochromes, all of them are confined in the GAF domain [30]. Conversely, PAS and GAF domains are known to be conserved domains among fungi and bacteria [81] and are involved in the stabilization of the chromophore. In fungal phytochromes, additional regulatory domains have been observed at the C-terminus [26]. Other differences with respect to the plant analogous are the maintenance of the proton activated/deactivated histidine kinase functions and D-ring mechanisms oriented in α-facial position in bacteria.

In BphP, BV is covalently attached via the C32 linkage of the terminal vinyl group; despite the covalent bond not being mandatory for photoconversion as the evidence suggests [82,83], the covalent bond provides stability to the holoprotein in the photoswitching process [80]. Regardless of the stabilization of the chromophore, a knot conformation has been observed at the interface of the PAS and GAF domains, containing an Arg254 that stabilizes the B-ring carboxylate of the BV. Mutations at the interface between the PAS domain and the knot, as well as mutations in the back side of the helices of the GAF domain, have been related to loss/gain in function [80]. Similarly, partial negative charges from His and Asp residues stabilize the protonated BV chromophore. It has been observed that the A-ring of the BV is tight in places, surrounded by the GAF domain and secured by the C10 methionine bridge and thioester linkage, avoiding bond rotation due to sterically induced effects. Therefore, the D-ring from the chromophore, whose chemical environment is looser, is strongly related to the photoswitching behavior and Pr-Pfr conversion [80].

Photoswitching process in BphP has been tackled through time-resolved serial femtosecond X-ray diffraction [84], pump-probe spectroscopy [84], femtosecond stimulated Raman spectroscopy [85], time-resolved wide-angle X-ray scattering [86], transient infrared spectroscopy [87], circular dichroism [81,88], and quantum mechanics/molecular mechanics (QM/MM) simulations [89,90]. Knowledge of the process suggests a counterclockwise chromophore isomerization. QM/MM simulations refer to a rotation of the τC (C-ring) and τD (D-ring) following the double and single bonds rotation in the Hula Twist mechanism and in accordance with the movable D-ring of the chromophore. The modification in the arrangement of the chromophore starts with a ZZZssa (C5-Z, C10-Z, C15-Z, C5-syn, C10-syn, C15-anti) conformer and achieves a ZZEssa (C5-Z, C10-Z, C15-E, C5-syn, C10-syn, C15-anti) after the excitation, restoring the ZZZssa configuration in most cases according to the QM/MM simulations. During the recovery and after the excitation state, the D-ring shows a perpendicular position with respect to the rest of the chromophore, that conformation is stabilized by a hydrogen bond network between the carboxyl and amino groups from the cofactor and buried water molecules. Further, a more planar and recovered position is achieved after the donation of a hydrogen bond with the hydroxyl group of Tyr263.

### 4.2. Photoswitching Mechanism of LOV Domain Proteins

LOV domains are part of the superfamily of PAS domains, which is conserved among eukaryotic and prokaryotic organisms. These proteins are involved in sensing light, oxygen, and voltage [91]. LOV proteins follow the basic structure of other signaling molecules in the presence of an OPM for biological functions; they are also related to the GAF sensor domain [23]. In accordance with the general structure of PAS domains and phototropins, LOV domains are constituted by a five-stranded antiparallel β sheet, 2–4 helical connectors, and an amphiphilic helix at the C-terminus called Jα [23,91]. However, LOV domains absorb blue light since they contain a flavin chromophore (FAD/FMN) (Figure 2B), which, in contrast to BphP, is not covalently bonded to the protein domain. Furthermore, it has been reported that a linkage with a cysteine present at 3.9 Å [22] in the pocket is produced by illumination with blue light, exciting the chromophore to a singlet state [92]. This flavin-cysteine adduct stimulates conformational changes in the tertiary structure of the LOV domain, triggering biochemical signaling outputs [93,94]. Despite this phenomenon, it has been observed that rather than being relevant in the linkage between the chromophore and the protein, the cysteine residue promotes the reversibility of the photoswitching [23]. In general, LOV domains modulate the activity of target proteins by inducing structural changes in the outward region of the LOV core when exposed to light. The versatility of these proteins is also attributed to their compositional characteristics, since LOV domains are soluble and exhibit sizes ranging ~100–140 amino acids. Notably, their light-sensitive cofactor is ubiquitously present across various cell types [91].

Regarding their mechanism, fungal and bacterial LOV domains have been distinguished due to their exceptionally long lifetimes of photocycle ranging from hours to days, where lifetimes in the order of seconds are the usual [95]. To explain these observations, stabilization effects have been proposed according to experimental data and simulations. The general mechanism begins with the formation of the Cys-FMN adduct after blue light illumination, which, in turn, produces the protonation of the N5 in the chromophore, which is subsequently attacked by the cysteine. This reaction has been monitored through the absorbance at 660 nm (absorption of the chromophore at the triple state) and 390 nm (absorption of the Cys-FMN adduct) [96]. Upon the formation of the adduct in the LOV2 domain of *Avena sativa* L., the existing interaction between Jα and the solvent-exposed β-sheet is destabilized [23,97]. Nonetheless, in the bacterial LOV domain of Bacillus subtilis, this interaction has been discarded because the helix is more planar and the connecting loop with the LOV core is shorter, making the interaction between the helix and the β-scaffold difficult [23]. In fungi and bacteria, a reorientation of a Gln in the β-sheet 5 was observed, in addition to a coordinate alteration of a N-terminal element at the external PAS core called Ncap, that is composed of a α-helix, a β-sheet, and a short hinge that connect the terminus with the core of the LOV domain; specifically in the VVD, the reorientation of the Ncap and Gln induces the dimerization of the LOV protein [23,95]. These findings regarding LOV domains in plants, fungi, and bacteria occur without significant changes in the overall structure of the LOV core, promoting crucial structural configurations to enable signaling during switching. Finally, strong hydrogen bonding and well-protected amines have been observed after irradiation, which produces a very slow exchange [22]. Conversely, the rate-limiting step in the dark conversion has been associated with the deprotonation of the N5 position from the FMN, which has been supported by experiments with imidazole producing a decay in the conversion rates of several orders of magnitude in AsLOV [95,98]. Two regions have been reported to be associated with the stabilization of the adduct in VVD; site 1 (Ile74, Cys76, Thr83, and Ile85) affects the stability of the cysteine in the adduct and regulates solvent/base accessibility to the flavin active site; residues at site 2 (Met135 and Met165) interact with the reface of the flavin ring; mutation at these residues has dramatic effects on recovery [95]. Through this whole series of effectors and interactions, the residues in sites 1 and 2 modulate the steric and electronic environment of the cofactor, stabilizing the adduct up to lifetimes in the order of days.

### 4.3. Photoswitching Mechanism of BLUF Domain Proteins

BLUF domains seem to exhibit similar mechanistic characteristics with LOV domains since alterations at the interface of the β-scaffold appear to mediate signal transmission; nonetheless, LOV and BLUF domains are remarkably distinct and follow different photochemistry mechanisms [91]. The general structure of BLUF domains is composed of two α-helices arranged adjacent to a β-scaffold (Figure 2B) [99]. Unlike BphPs and LOV domain, where the chromophore undergoes photoisomerization and the formation of an adduct, respectively. The switching behavior of the BLUF domain remains in the hydrogen bond network [91,100]. In this regard, Tyr21, Trp104, Met106, and Gln63 (numeration from Appa BLUF) are the residues involved in the hydrogen bonding at the chromophore pocket. Additionally, BLUF proteins exhibit photoinduced proton-coupled electron transfer [101]. There are discrepancies between the positions of Trp and Met at the ground state of BLUF domains, whereas in some crystal structures, the Trp of the pocket seems to be towards the flavin (Trp in) chromophore and the Met faces away from it (Met out); other models present the inverted conformation. However, these disparities have been examined through QM/MM simulations, linking the different conformations to the equilibrium at the ground state [101]. Additionally, Goyal and coworkers associated the multiexponential decay rates with this heterogeneity in the ground state. Nonetheless, the Trp out/Met in conformation seems to be well oriented to promote the proton relay. 

Simulations indicate that, after photoexcitation, the cofactor achieves an excited singlet state (locally excited state) that induces electron transfer, followed by proton transfer [102]. A proposed mechanism by Mathes and coworkers [103] suggests an electron transfer from the Tyr to the cofactor after achieving the locally excited state and then a proton transfer from the same Tyr to the cofactor. However, in a study using the Appa BLUF photoreceptor, a reorientation of the OH group of Tyr toward Gln and a rotation of Gln toward the cofactor has been observed, possibly associated with the proton transfer from Try to Gln and a consecutive proton transfer from Gln to the flavin [101]. Similarly, a study comprising the Slr1694 BLUF protein [54], indicates a more favorable interaction between the Tyr and Gln residues than a direct interaction with the Glu residue. Additionally, the authors provide evidence that the formation of a charge transfer state precedes the proton transfer, which is higher in energy than the locally excited state. Nonetheless, this charge transfer state can be reversed to the locally excited state if the proper conditions of electrostatic environment and donor-acceptor distance are not satisfied. Moreover, through ultrafast transient spectroscopy, a red intermediate was observed in PixD and PapB BLUF domains [100], where Tyr and Gln residues aid in the formation of the intermediate; nevertheless, this intermediate has not been reported in the Appa photoreceptor. Throughout the observations of ultrafast transient spectroscopy, the authors suggest a ketoenol tautomerism in the Gln side chain during the formation of the red intermediate and observe strong hydrogen bonding with Tyr.

Although the function of photoswitchable proteins in plants is extensive, the general mechanisms have not been elucidated, especially for nonphotosynthetic bacteria, where the biological function of these proteins remains unclear for most of the systems [30]. The vast repertory of OPM allows a specific interaction between photosensory modules and suitable effectors, providing the basis for a complete optogenetic platform. Furthermore, these proteins can be engineered through rational design, leading to either enhancement or reduction in their activity. Consequently, in-depth knowledge about the mechanistic and structural aspects of photoswitchable proteins permits researchers to fine-tune their optokinetic properties, expanding their potential applications.

## 5. Novel Applications

### 5.1. NIR-Based Imaging

Despite the fact that fluorescent proteins (FPs) from jellyfish and corals have revolutionized the optical imaging of cells, they only fluoresce in the visible wavelength range [104]. Due to their absorption by hemoglobin, water, and lipids, these proteins are not appropriate for deep-tissue imaging. The excitation and emission maxima of FPs have not yet exceeded 598 and 655 nm, respectively. A highly desirable feature of in vivo imaging techniques is being able to use FPs in the so-called near infrared optical window (from 650 to 900 nm), because the mammalian tissue is more transparent to light due to the combined absorption of hemoglobin, melanin, and water is minimal [105]. Interestingly, longer wavelengths (644-nm excitation, 672-nm emission) have been observed in bacterial phytochromes (BphPs) that incorporate phycocyanobilin (PCB) as a chromophore. It has been possible to modify BphPs to make them more efficient in the infrared range in animal models. BphPs have the advantage of easy modification due to their modular structure; also, their absorption spectrum is the most red-shifted among the phytochromes, and there is the possibility to use biliverdin IVα (BV) as a chromophore. BV is a red-shifted natural chromophore and an intermediate heme metabolism, and it is ubiquitously found in mammalian tissues. For these reasons, BphPs are excellent templates for the engineering of near-infrared fluorescent proteins (NIR-FPs). Promising NIR-FPs (Table 2) have been developed as permanently fluorescent proteins (IFP1.4, Wi-Phy, IFP2.0, IFP1.4rev, iRFP670, iRFP682, iRFP702, iRFP713, and iRFP720), as photoactivatable fluorescent proteins (PAiRFP1 and PAiRFP2), and reporters for protein-protein interaction (iSplit, iRFP BiFC system, IFP PCA, and fragmented IFP) [106,107]. Most of the mutations created in BphPs have the objective of stabilizing the chromophore Pr conformation by destabilizing the BV conformation or by decreasing the BV deprotonation rate and thus the likelihood of excited state proton transfer [108]. Several drawbacks of the first NIR-FPs that hindered their use as biomarkers included dimeric configuration, low fluorescence brightness in mammalian cells, and exogenous BV cofactor dependency. The development of bright monomeric spectrally NIR-FPs made them more suitable to be used as in vivo biomarkers.

Improvements are still being developed to optimize the properties of NIR-FPs. The new and smaller version miRFP670nano3 with a size of 17 kDa brightly fluoresces in mammalian cells, enables deep-brain imaging, and is an excellent component for NIR fluorescent nanobodies (Figure 3G), NIR-Fbs [117]. NIR-Fbs allowed the visualization of endogenous proteins, identification of viral antigens, labeling of cells expressing specific molecules, and identification of double-positive cell populations with bispecific NIR-Fbs against two antigens [117]. From these NIR-Fbs, many molecular tools can be developed for the directed degradation, expression or modulation of targeted proteins, and the manipulation of a variety of cellular processes based on the intracellular protein profile.

Noninvasive monitoring in deep tissue that allows tracking and visualization requires both new imaging techniques and advanced probes. NIR-FPs are useful for imaging cells, issues, or whole organisms, as we observed in Figure 3 and Table 2. NIR-FPS enables more advanced in vivo imaging at greater resolution than previously achievable and the possibility of monitoring several more tags in individual cells or tissues (Figure 3A). NIR-FPs proteins have made multicolor imaging possible to follow up on the mechanism of metastasis in cancer cells (Figure 3I). Metastatic dissemination and tumor growth were monitored in vivo through NIR optical imaging with a lentiviral vector to deliver a NIR-FPs gene into different tumor cell models [118]. The use of mouse xenograft models that are based on the implantation of human tumor cells into immunocompromised mice has also been used to demonstrate the stability and function of NIR-FPs (Figure 3D) [116]. An interesting modality for the use of NIR proteins in live imaging assays is their packaging into VLPs (viruses, such as particles). With this configuration, VLPs exhibited stable photochemical properties for noninvasive in vivo imaging in mice (Figure 3C) [119]. 

For in vivo applications, such as cell-based therapy, NIR-FPs can help track engrafted cell survival and biodistribution (Figure 3B). Engrafted stem cells that were modified to express iRFP reporter genes through lentiviral labeling showed excellent monitoring of viability after transplantation [120]. Interestingly, retinal neurons tagged with iRFP by an adenovirus were a noninvasive marker for physiological identification without interfering with the recording of photoreceptor-mediated light synaptic responses (Figure 3F) [121].

On the other hand, NIR-FPs proteins have allowed the separation of populations of cells by flow cytometry using visible red and NIR lasers (Figure 3E) [122]. Since NIR-FPs have a wavelength that penetrates the tissue, they are well-suited to optical tomographic reconstruction, photoacoustic tomography, and photoacoustic microscopy (Figure 3H). Photoacoustic (PA) imaging consists of the use of a combination of optical excitation and acoustic detection to overcome the traditional depth limitations of optical imaging. A promising protein for monitoring tumor growth and metastasis is the nonfluorescent bacteriophytochrome RpBphP [114]. Nevertheless, the protein DrBphP has been described as having better performance than its split variant, DrSplit [123]. However, one of the main challenges for the use of this type of protein in diagnostic methods is that its photostability remains low because of photobleaching and transient absorption.

### 5.2. Optogenetics

The term “optogenetics” was used to define the selective expression or inhibition of genetically targeted photoreceptor expression in neurons by light. Subsequently, this term was extended to cover all types of applications of genetically encoded photosensitive proteins. Light is an excellent alternative to chemical inducers used to activate genetic circuits due to its low cost, reduced toxic effect, adjustable level, and higher level of expression [124]. Through optogenetics, it has been possible to control cellular functions such as transgene activation, genome activation/editing, cell migration, disruption of signaling pathways, modulation of customized phenotypes, and stimulation or inhibition of metabolic pathways. The genetic circuits are activated by illumination with a specific wavelength, and consequently, the domains that are sensitive to light undergo a conformational change. This can result in proteins either dimerizing or dissociating from other proteins or domains, as well as causing partial folding/unfolding of the protein structure [43]. These changes subsequently exert control over diverse biological processes through the recruitment of transcriptional repressors or activators, as well as through the regulation of translation [125].

The photoreactive proteins used mainly come from plants (phytochromes, cryptochromes, LOV, UVR8), but also from bacteria, archaea, algae, and higher animals (rhodopsins, cyanobacteriochromes, cryptochromes, phytochromes, and LOV domain proteins). Additionally, those proteins can be divided into channel proteins (rhodopsins, channelrhodopsins, and halorhodopsins) and intracellular proteins (LOV, BLUF, CRYs, phytochrome, and UVR8 families) [126]. Bacterial photoreceptors have been widely used, while the potential of fungal photoreceptors as biological components for constructing optogenetic switches has not been fully utilized. 

The dimeric or oligomeric states of certain proteins that respond to light play a crucial role in controlling signaling and transcriptional processes. Some optogenetic switches are based on self-dimerization, through a LOV (light, oxygen, and voltage) domain. The most common fungal domains are from the VVD, the smaller protein containing the oxygen light voltage LOV domain) [126], and from White Collar-1 (WC-1), both from Neurospora crassa. In the case of bacterial origin LOV modules, among the most used are the blue-light photoreceptor called EL222 from Erythrobacter litoralis, which binds DNA when illuminated with blue light [127]. Additionally, the protein RsLOV from Rhodobacter sphaeroides exists as a homodimer in the dark and as monomers after blue light-induced dissociation. This behavior can be employed to manipulate the function of an effector domain by hindering it through steric obstruction [127]. 

Bacterial phytochromes (BphPs) have several advantages for optogenetic control; they carry enzymatically active effector modules, they are assembled in the configuration of two-component signaling systems, and they are activated with far-red or NIR light. The main proteins used in this class are BphP from *Deinococcus radiodurans*, BphP1 from *Rhodopseudomonas palustris*, and BphG1 from *Rhodobacter sphaeroides*. BphP acts as a phosphatase, while its photosensitive component regulates the histidine kinase function of similar receptors that promote subsequent gene expression [128]. The light-responsive mechanism established through the reversible binding of the bacterial phytochrome BphP1 to its inherent partner PpsR2 can be used to induce several types of cellular processes (recruitment of proteins to specific cellular locations, initiating internal enzymatic reactions, and activation of gene expression) [128]. On the other hand, the synthesis of second messengers is a regulatory modality that has been successfully placed under light control to express specific genes through BphG1. The bacteriophytochrome diguanylate cyclase, derived from the Rhodobacter sphaeroides BphG1 protein, generates cyclic dimeric GMP (c-di-GMP) from GTP when exposed to light. This process is exclusive to certain organisms and is absent in humans, making it a promising distinct controller of gene expression [129].

Many genetic circuits have been developed by combining these light-responsive domains. Through the presence of a certain type of light, its absence, or its combination with another type of light, it is possible to control cellular processes, which can also have a reversible effect. For example, the change of violet/green, blue/dark, green/red, red/dark, red/NIR, and blue/green, among others, has been observed [130]. However, extremely high intensities of certain types of light can affect cell growth. In the case of yeasts, high-intensity blue, green, and white light significantly reduces their growth [131]. Several systems have been constructed to optimize the control of expression in yeast, such as the optogenetic switch called FUN-LOV (FUNgal light oxygen and voltage) for yeast protein expression [124,132]. On the other hand, in human tissues, blue light has poor penetration and phototoxicity. For that reason, NIR light is a promising tool for optogenetic-based therapy in regenerative medicine. NIR control is suitable for using CRISPR-dCas9 for transcriptional activation in the induction of the differentiation of induced pluripotent stem cells into neurons. This mutant form of Cas9 (dCas), whose endonuclease activity was removed, is commonly used to repress transcription factors in the promoter of the gene of interest. The light sensor consisted of the bacterial phytochrome BphS, which converts GTP into c-di-GMP upon light illumination (730 nm), and then c-di-GMP induces the dimerization of transcription factors that will cause the overexpression of specific neural transcription factors [133]. 

Optogenetics offers the potential to manage biological processes with exceptional precision in both timing and location, causing minimal disruption. With advances in massively parallel sequencing technologies, new fungal and bacterial proteins can be identified and used to develop new optogenetic tools. In addition, promising results from various study areas suggest that optogenetic tools are an option for new therapies, control systems, diagnosis, and bioproduction, among others.

### 5.3. Optoprotein-Based Materials

Optoprotein-based materials (OPMs) refer to a class of materials functionalized with photoswitchable proteins [134]. This modification allows the manipulation of material properties, including mechanical behavior [135], thermal conductivity [136], cell adhesion [137], optical response [138], conductivity [139], and morphology [140]. In the scientific literature [141,142], there are references to “smart” materials that show sensitivity to changes in light; however, it must be considered that most of these resulting alterations are usually irreversible. Unlike other materials with photosensitivity, OPMs possess a distinctive feature: their reactions can be reversed when exposed to darkness or a specific wavelength [134]. Furthermore, they also show specific interactions with different wavelengths of light. In practice, glass meshes [137,139], hydrogels [143,144], and aggregated colloidal particles represent typical materials effectively functionalized using optoproteins [145]. The mechanisms of light-induced assembly, including homodimerization, heterodimerization, and oligomerization, are depicted in Figure 4. These processes play a crucial role in altering the degree and quantity of crosslinking and adhesion when photoswitchable proteins are incorporated within materials [134].

Hydrogels are one of the most commonly used materials for incorporating photoswitchable proteins to modify properties [135,146,147]. These soft materials are composed of natural or synthetic polymers and are popular due to their ability to store water and their biocompatibility when in contact with biological surfaces [148]. Thanks to their remarkable ability to maintain contact with living tissue, functionalized hydrogels with photoswitchable proteins are a promising approach for regulating reversible cell adhesion in specific areas and time intervals [146,147]. Recent studies conducted by Yüz et al. [137] and Wang et al. [147] demonstrate how the incorporation of light-sensitive proteins into these stimuliresponsive materials enables them to mimic the dynamic cellular environment. Consequently, photoswitchable hydrogels represent an excellent opportunity to advance tissue engineering techniques by providing unprecedented precision in modulating cell attachment and detachment using a noninvasive tool, such as light. Figure 4A shows a graphical description of the use of functionalized films with photoswitchable proteins to manipulate cellular adhesion.

Photoswitchable proteins in films have been found to serve another purpose—the development of living materials, as reported by Sankaran et al. [149]. The study discovered that immobilizing bacteria, such as *Escherichia coli* can establish a dynamic interaction between this material and intricate mammalian cells on different surfaces. Through the optoregulation principle, new materials, such as bacteria-functionalized hydrogels, have been created that can selectively release adhesive molecules upon exposure to light. Furthermore, research conducted by Sankaran et al. [149] demonstrates that bacteria can efficiently transfer fluorescent molecules to mammalian cells when immobilized or functionalized on the surface of Nexterion glass. This innovative approach introduces a novel strategy for delivering molecules to mammalian cells using bacterial systems, supporting the growing need to develop advanced materials that provide process reversibility and the ability to signal living cells. The fabrication of dynamic materials with the regulation of signaling molecules that respond to stimuli such as light represents something new in the scientific field. The highlights of light activation are noninvasive, and self-regulating, and the response depends on the intensity and dose of the activating light.

Photoswitchable proteins have also been incorporated into hydrogel designs to create intelligent materials capable of responding to external stimuli and modifying their crosslinking behavior in response to specific wavelengths [143], as shown in Figure 4B. This phenomenon markedly impacts the scaffold’s rigidity and resistance against deformation because of dimerization events involving the photoreceptors integrated within the polymeric network [150]. Conversely, when exposed to a different wavelength, causing dissociation of these photoreceptors, there is a drop in crosslinking density, which consequently yields softening of the material [151]. This innovative technique has garnered significant attention in recent studies conducted by Hörner et al. [143], and Wang et al. [147], wherein photoswitchable proteins were employed for enhancing mechanical properties and rheological behavior across alginate- and polyacrylamide-based hydrogels that had undergone functionalization processes. Such advanced materials exhibit notable capabilities, including compression resistance or transforming from liquid states into semi-solid or solid forms depending on light exposure conditions, alongside the presence of certain cofactors [143,145].

The study of natural processes related to embryonic development, cell communication, and cell grouping has always been a significant area of research. Scientists have extensively investigated the intricate mechanisms involved in these processes to better understand how organisms develop from embryos and how cells interact with each other [152,153]. Self-assembling materials functionalized with photoswitchable proteins have been extensively investigated as a means of replicating these complex biological processes. As an example of these materials, functionalized colloidal particles have been developed to self-sort and undergo reversible assembly when exposed to specific wavelengths. This approach, inspired by nature’s ability to self-organize, holds great promise in many fields, such as drug delivery systems, sensors, nanotechnology, and tissue engineering. As seen in Figure 4C, Sentürk et al. [145] mention the development of colloidal nickel particles functionalized by interaction with histidine tags with two photoswitchable proteins, Cph1 from *Cyanobactrium Synechocystis*, and VVD from *Neurospora crassa* [154]. This functionalization confers on these particles the ability to self-assemble and group selectively and discriminately under exposure to blue and red light. The prospect of future applications of this type of self-assembling material is that biocompatible materials can be composed of aggregated cells and organized in different clusters under a non-invasive and specific stimulus such as light [145]. This approach offers valuable information on the fundamental mechanisms that govern self-organization throughout the different stages of development, expanding our understanding of this field. The surfaces presented here primarily intend to replicate the phenomena observed in tissues and their interactions with more complex cells. The innate ability of cells to self-assemble and sort into tissue-like architectures reveals great potential for bottom-up tissue engineering; tracking adhesions between cells is a powerful tool for programming synthetic tissues.

Additionally, new biomaterials that exhibit light sensitivity have recently emerged, utilizing photoactive yellow proteins (PYPs) to create integrated optics (IO) circuits for enhanced data transfer and processing. PYPs demonstrate a rapid photocycle in solution and display significant changes in refractive index when exposed to light in dried films [42]. Additionally, thin films containing oriented bacteriorhodopsins offer high stability and can be used alongside integrated optical devices [155]. The combination of the photonic structure of doped porous silicon microcavities with the photochromic properties of PYP proteins has shown promise in inducing light-induced reflectance changes. This innovative material holds potential as an alternative option for future incorporation into porous matrices compatible with integrated optics [156].

Table 3 showcases a selection of bacterial and fungal optoproteins employed for material functionalization. Notably, while both bacteria- and fungi-derived optoproteins have found utility in this field, the utilization of fungal variants remains comparatively limited. Yet, it is anticipated that their application will expand significantly in the future. Additionally, an examination of Table 3 reveals a broad spectrum of stimulating wavelengths, providing ample flexibility for substrate functionalization methodologies such as click chemistry, spy chemistry, or interactions with histidine tags.

### 5.4. Bioluminescence to Detect Pollutants

Bioluminescence is a chemical heatless reaction made by some living higher and microorganisms that produces visible light. Several other enzymes, in addition to luciferase, contribute to the process of bioluminescence by creating luciferin from cellular metabolites and converting it back to its reduced form and in a generalized way are known as bioluminescent proteins. Typical application of the bioluminescence falls in the cellular biology field to trace complex metabolic pathways and physiological regulatory elements, in the gene expression field to corroborate the correct insertion of DNA sequences, protein-protein interactions and tracking the course of infection for the analysis of diseases. All of these applications of bioluminescence are complex areas in which it has been done an important effort in the development of tools and protocols, however, the use of the bioluminescence response of some microorganisms as a biosensor to detect and track pollutants in several ecosystems constitutes the simplest application of the bioluminescence and photoproteins and it has been growing and becoming an important and powerful technique [159]. The typical protocol consists in the measurement of the light emission reduction of the bioluminescent microorganisms in presence of a given concentration of a pollutant. Some pollutants that can be successfully detected by bioluminescence assays are polycyclic aromatic hydrocarbons, antibiotics, organic solvents, and heavy metals [159,160,161]. The presence of antibiotics in aquatic systems can be in the magnitude order of nanograms to some few micrograms, therefore for its detection and quantification is often necessary the use of expensive analytical techniques such as Ultra-High-Performance Liquid Chromatography (UHPLC) coupled to mass spectrometry (MS) or tandem mass spectrometry (MS/MS). An alternative to these expensive and time-consuming analytical techniques is the measurement of bioluminescence. For instance, Ioele et al. [160] and Steevens et al. [161] measured the bioluminescence decrease of *Vibrio fischeri* in presence of several antibiotics ranging from 1 to 50 μg/mL such as Chlortetracycline, Oxytetracycline, Ofloxacin, Norfloxacin, Oxolinic Acid, Nalidixic Acid and Streptomycin. The bioluminescent microorganism of the genus *Vibrio* sp. has been preferred during this kind of study. However, two important concerns arise when this kind of protocol is used, (i) the bioavailability of these toxic compounds and (ii) the toxicity level in higher organisms. Nevertheless, this field continues to be under study for the development of robust and efficient biomarkers.

### 5.5. Intensification of Bioprocesses Based on the Optocontrol

One of the challenges in current biotechnology, where a given microorganism is the responsible to produce the desired biomolecule, is to achieve a control of the product formation or the substrate to product ratio formation which is of interest to control the bioprocesses to increase productivity [13]. The key for controlling and increasing the productivity in a bioprocess relies on the capability to keep the system under an equilibrium between biomass growth and the production of any biomolecule [1]. This kind of strategy will lead to a bioprocess that can be efficiently controlled by artificial intelligence instead of traditional control based on the sample handling and monitoring process (Table 4) [13]. When microorganisms are used to produce a given biomolecule, there is the typical production associated to the growth or the non-associated to the growth, in both cases a basic control other than the operating parameters of the bioreactor can be done (Figure 5A). Then, an effort was conducted to the use of strategies such as those based on a two stage bioprocess where a simple carbon source such as glucose was used to increase the biomass concentration in a growth stage and then, a second carbon source, often a complex carbon source, is used to produce the desired biomolecule in a production stage where a high concentration of biomass was accumulated during the previous stage and therefore high conversion rates can be achieve (Figure 5B). For instance, the production of biopolymer (polyhydroxyalkanoates) or aroma compounds (lactones) typically employed the strategy of two-stage process [162,163]. Finally, there is the genetic control in which chemical inducers, such as the isopropil-β-D-thiogalactopyranoside (IPTG) for recombinant *Escherichia coli*, has been used to control the genetic expression of a desired biomolecule and to achieve control over the change from a growth stage to a production stage (Figure 5C). However, the use of chemical inducers increases the cost of the bioprocess as well as having a toxic effect or disrupting some of the downstream processes. Moreover, the introduction of a chemical inducer in the culture medium provokes continuous and irreversible gene expression that may induce a cell stress that does not always end in high productivity of the bioprocess [13,164]. Lately, the development of a new control technology for genetic expression through the light can offer a better balance between growth and production stages in order to increase productivity. This technology offers the control of the bioprocess using the presence/absence of a specific wavelength of light, which is less invasive and can be effectively modulated in time and intensity, also it is a reversible induction since the light can be turned off/on depending on the optocontrol allowing a dynamic control alternating between growth and production phases, which can be related to a kind of photoperiod (Figure 5D). 

This dynamic control offers a “rest” to the cell for recovering from the production phase and the productivity of the bioprocess can be enhanced as was demonstrated by Zhao et al. [164] the production of isobutanol by transformed *Saccharomyces cerevisiae*. The use of this strategy would allow a better balance between growth and bioproduction, and consequently increasing the control of the bioprocess while achieving higher productivity. Moreover, Pouzet et al., [165] the intensification of a bioprocess through the optocontrol under an approach of cybergenetic control in which the growth/production phases can be controlled given the physiological state of the cell. Several studies on different microorganisms have tested the idea of an optocontrol based on the optogenetics; and microorganisms such as *E. coli*, *Saccharomyces cerevisiae, Pseudomonas putida, Pichia pastoris,* and *Bacillus subtilis* has been successfully modified to implement an optosystem for control purposes. Biomolecules such as mevalonate, lycopene, isobutanol, among others have been successfully produced in these engineered microorganisms by using a specific wavelength of light to modulate the production. An interesting application of photocontrol was proposed by Salinas et al. [124] the biomass removal after a fermentation process. They proposed the use of the light-oxygen-voltage (LOV) domain to control the flocculation process in the yeast Saccharomyces cerevisiae and therefore to design a more efficient and fast way to separate the cells from the culture media. Another interesting function of the LOV domain relies on the study of the pathway for carotenoid production of the fungus *Blakeslea trispora* (Luo). Although little is known about the light induction of carotenogenesis in *B. trispora*, there is a potential interest in clarifying this pathway to increase the carotenoid productivity of this class of fungi, which is recognized to have an important role in large-scale production of carotenoids.

**Table 4 ijms-24-14741-t004:** Novel applications of the optocontrol strategy in several bioprocesses.

Bioprocess to Control	Micro-Organism	Optogenetic System	Importance	Ref.
Mevalonate and isobutanol production at 2 L bioreactor scale	*Escherichia coli*	OptoLAC circuit based on the control of lacI by the PR promoter of the pDawn system. The blue light was used to control the process.	Production was significantly increased compared with a traditional IPTG system of genetic control. The OptoLAC circuit can be used in other applications for E. coli currently based on IPTG induction.	[166]
The programmable assembly/disassembly of membraneless organelles	*Saccharomyces cerevisiae*	PixELLs optocontrol system	The control of engineered metabolic pathways by assembly and disassembly of metabolically active enzymes clusters.	[167]
Ethanol and isobutanol bioproduction	*Saccharomyces cerevisiae*	OptoINVRT-ILV2 (Ethanol) and OptoEXPPDC1 (Isobutanol). The blue light was used to control the process.	It offers an option to delete essential pathways that compete with the pathway of the interest bioproduct.	[164]
Induction of the autoflocculation process of yeast.	*Saccharomyces cerevisiae*	Light control of the flocculin-encoding gene FLO1 by the FUN-LOV optocontrol system	It offers a low-cost alternative to typical bioseparation performed by membranes.	[124]

## 6. Conclusions

The study of optoproteins has been found to be crucial in advancing a variety of scientific fields of interest, including the regulation and control of bioprocesses, cell biology and the development of new materials. Particularly, bioluminescence was identified as a valuable tool in gaining a deeper understanding of intracellular processes, with potential applications in accurately monitoring harmful substances in the environment and quantitatively evaluating the progress of diseases. However, the potential of photoswitchable protein functionalization for creating innovative materials and improving bioprocess control methods to reduce costs and improve productivity has yet to be fully explored. Additionally, it was noticed that most reported applications have remained at the laboratory level, highlighting a significant opportunity area for sustainable technological development. Future work should focus on developing clear proposals to industrialize these novel applications using optoproteins, so it is necessary to continue with the research and technological development of high-value proposals with development and scaling capabilities.

## Figures and Tables

**Figure 1 ijms-24-14741-f001:**
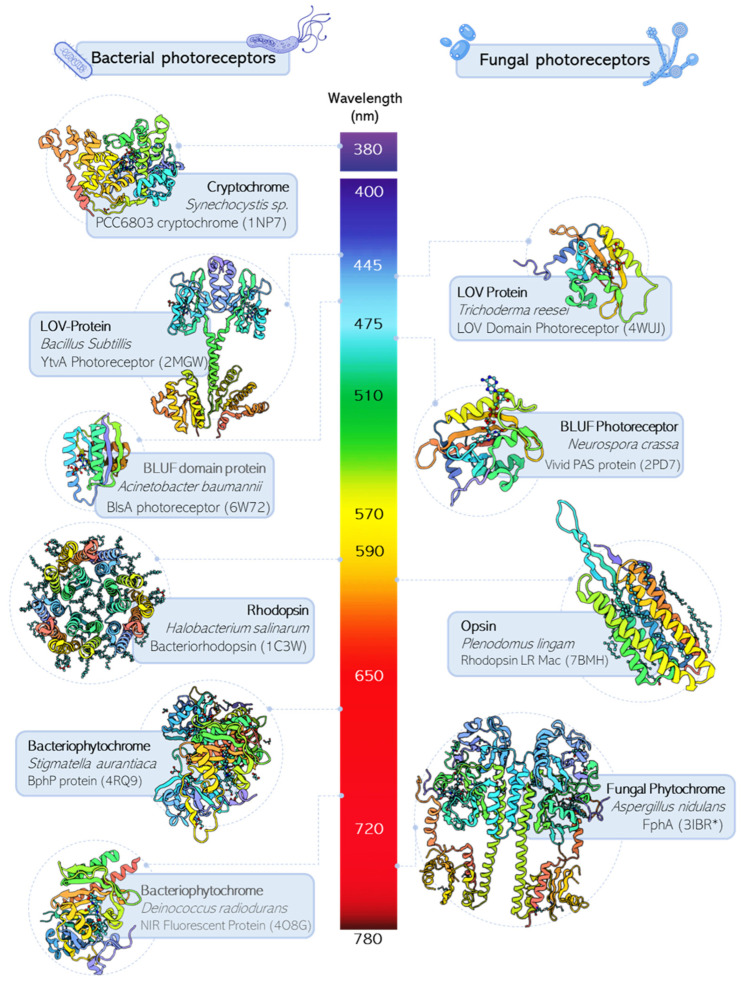
Bacterial and fungal photoreceptors. Many of the photoreceptors of fungal and bacterial origin have been abundantly analyzed, and their crystallographic structures have been elucidated, which has allowed for a more exact understanding of their photoreceptive mechanisms. In the figure, you can see the structures of some types of photoreceptors, which are connected to the visible spectrum at the wavelength corresponding to their absorption capacity. The PDB code corresponding to each structure has been placed between parentheses. In the case of photoreceptors from fungi, it was observed that there is a lower abundance of elucidated crystallographic structures. * Such is the case for the red light-sensing photoreceptor FphA from *Aspergillus nidulans,* which is known, by sequence alignment, to only differ by 5 aminoacidic residues from the *Pseudomonas aeruginosa* photoreceptor. Partially created with BioRender.com.

**Figure 2 ijms-24-14741-f002:**
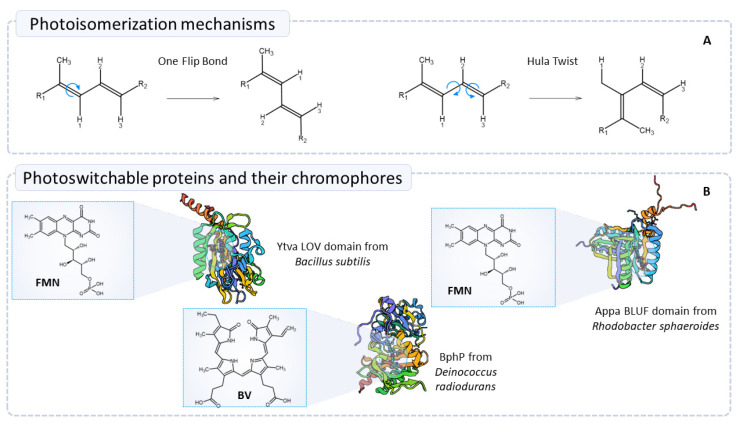
Photoisomerization mechanisms and cofactors of photoswitchable proteins. (**A**) The main mechanisms of photoisomerization. OFB mechanism produces a trans-S-trans conformation, meanwhile HT produces a trans-S-cis conformation. (**B**). Examples of the main photoswitching protein families found in fungi and bacteria; *left,* an example of LOV domain protein (PDB: 2PR5) with a flavin chromophore; *center,* an example of BhpP (PDB: 4O8G) with a biliverdin chromophore; *right,* an example of BLUF domain protein (PDB: 1YRX) with the flavin chromophore. Partially created with BioRender.com.

**Figure 3 ijms-24-14741-f003:**
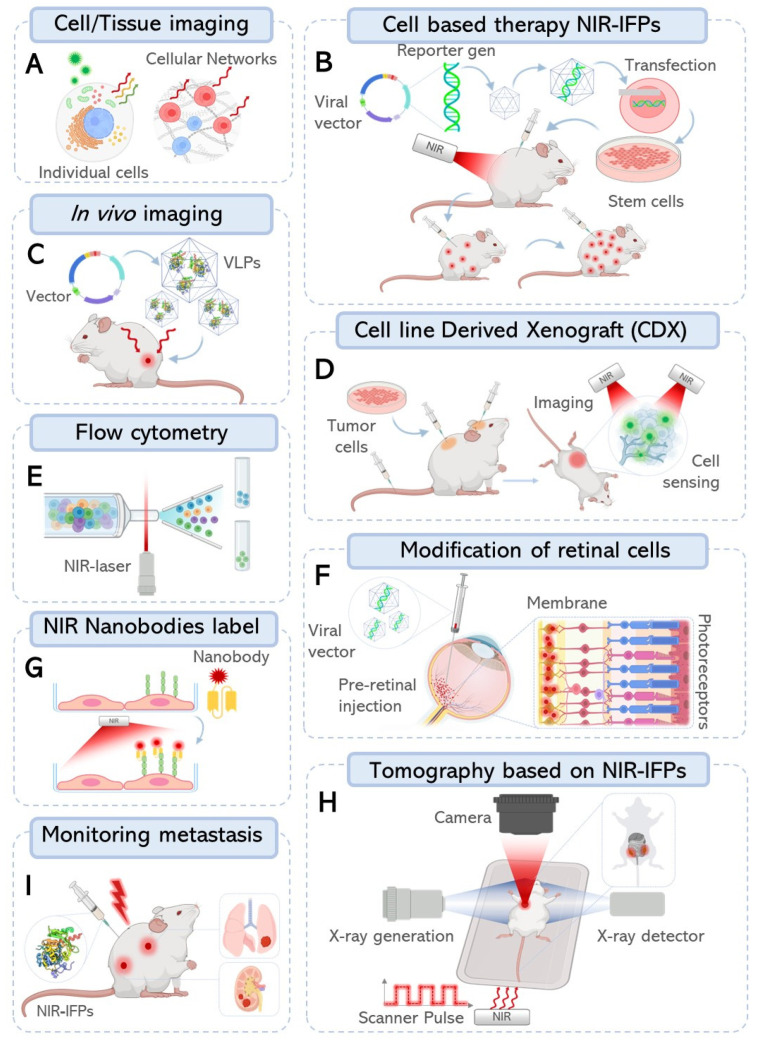
In vivo and in vitro NIR-based imaging applications with NIR-FPs. (**A**) Visualization of individual cells and cell network interactions. (**B**) Monitoring of modified stem cells that express NIR-PFs by a viral vector strategy. (**C**) Packaging of NIR-PFs into VLPs for in vivo imaging. (**D**) Stability and function in mouse xenograft models. (**E**) Separation of cell populations by flow cytometry. (**F**) Study of cell physiology without interference from retinal neural photoreceptors. (**G**) NIR fluorescent nanobodies for visualization of specific biomarkers. (**H**) Use of NIR-FPs to perform a tomography. (**I**) Cancer cells labeled with NIR-FPs make it possible to track metastasis in vivo. Partially created with BioRender.com.

**Figure 4 ijms-24-14741-f004:**
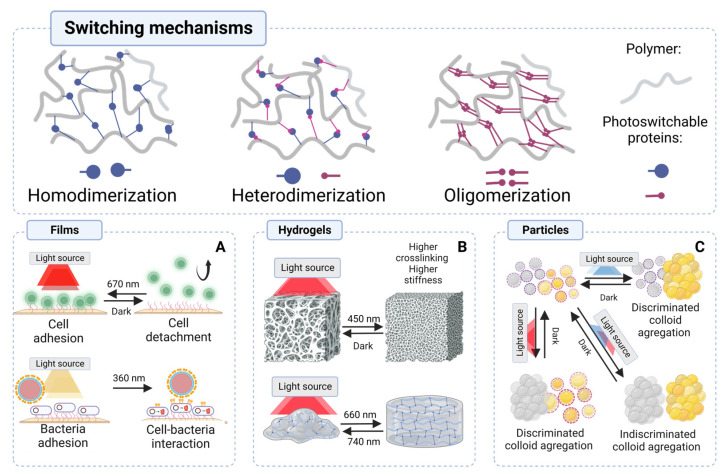
Examples of Optoprotein-based materials. (**A**) Functionalization of thin films and substrates with photoswitchable proteins to reversibly capture cells when exposed to a wavelength of 670 nm and detach them in the dark. (**B**) Functionalization of bulk hydrogels to improve stiffness and crosslinking at different wavelengths. As mentioned by Hörner et al. [143] This process can also be performed for the reversible formation of hydrogels from a liquid to a solid state by exposing the material to red light (660 nm). (**C**) Self-sorting self-assembly in a mixture of particles functionalized with different photoswitchable proteins. Sentürk et al. [145] mentioned that it is possible to disperse particles in the dark, aggregate them in a discriminant way under red or blue light, and indiscriminately aggregate them using coillumination with blue and red light. Partially created with BioRender.com.

**Figure 5 ijms-24-14741-f005:**
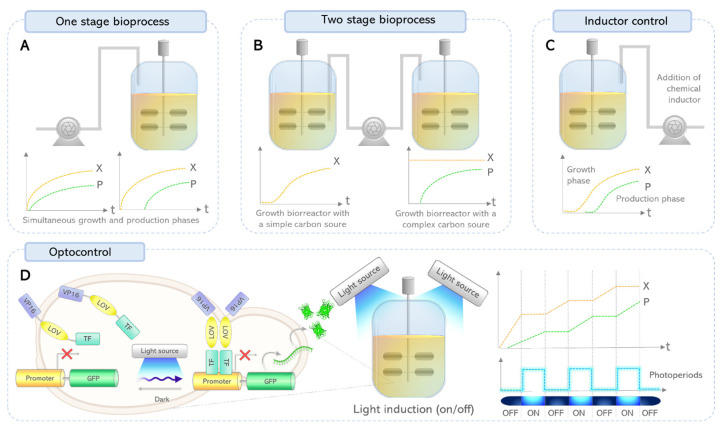
Bioprocess control strategies. (**A**) Typical bioprocess control strategy based on the measurement and control of operating variables and production associated with or not associated with the growth. (**B**) Bioprocess is based on the addition of different carbon sources to control the production stage. (**C**) Chemical control to modulate the expression of certain genes to start the production stage. (**D**) Optocontrol of the production stage based on the modulation of gene expression through photoperiods.

**Table 1 ijms-24-14741-t001:** Photoswitchable proteins from fungi and bacteria from the Optobase database [77].

Name	Kingdom	Chromophore	Excitation Wavelength	Reversion Wavelength	Type
Am1 c0023g2/BAm	bacteria	Phycocyanobilin	680 nm	525 nm	Cyanobacteriochrome
PixD/PixE	bacteria	FAD/FMN	450 nm	dark	BLUF
BcLOV4	fungi	FMN	450 nm	dark	LOV
EL222	bacteria	FMN	450 nm	dark	LOV
pMag/nMag	fungi	FAD	450 nm	dark	LOV
NcWC1-LOV	fungi	FAD	450 nm	dark	LOV
RsLOV	bacteria	FMN	450 nm	dark	LOV
VVD	fungi	FMN/FAD	450 nm	dark	LOV
YtvA	bacteria	FMN/FAD/RF	450 nm	dark	LOV
PYP	bacteria	p-coumaric acid	450 nm	dark	fluorescent
MxCBD	bacteria	AdoCbl, MetCbl or CNCbl	545 nm	dark	cobalamin-binding
TtCBD	bacteria	AdoCbl, MetCbl or CNCbl	545 nm	dark	cobalamin-binding
BphP1/PpsR2	bacteria	biliverdin	760 nm	640 nm/dark	phytochromes
BphP1/Q-PAS1	bacteria	biliverdin	760 nm	640 nm/dark	phytochromes
Cph1	bacteria	PCB	660 nm	740 nm	phytochromes
DrBphP	bacteria	biliverdin	660 nm	780 nm/dark	phytochromes
iLight	bacteria	biliverdin	660 nm	760 nm	phytochromes
MagRed	bacteria	biliverdin	660 nm	780 nm/dark	phytochromes

**Table 2 ijms-24-14741-t002:** Examples and characteristics of NIR-FPs developed for imaging.

NIR-FPs	Ex/Em (nm) and Organization	Properties	Bacterial Phytochrome Template	Ref.
IFP1.4	684/708 nm monomeric and dimeric.	It expresses well in mammalian cells and mice. It does not fluoresce without a BV supply. It is the brighter version of IFP2.0.	Designed from DrBphP (*Deinococcus radiodurans*). It has a truncation of the PHY and C-terminal histidine kinase–related domains and underwent mutagenesis.	[109]
Wi-Phy	701/719 nm monomeric	Its application in cells or animals has not been demonstrated.	A variant of the *Deinococcus radiodurans* DrCBD-D207H with Y263F	[108]
IFP1.4rev	685/708 nm	It presents high brightness and a large complementation contrast, but it is irreversible. Applicable for in vivo protein-protein interaction studies.	Arose from protein IFP1.4 with the mutation H207D.	[107]
iRFP713	690/713 nm dimeric	It is stable, noncytotoxic, and the low concentrations of endogenous BV are sufficient to make it brightly fluorescent in cells, tissues, and whole animals.	Generated from RpBphP2 from the photosynthetic bacterium *Rhodopseudomonas palustris*.	[110]
iSplit	690/713 nm separate domains	Configuration suitable as a reporter. Based on bimolecular fluorescence complementation (BiFC). Allows detection of interactions in vivo.	Produced by separating iRFP713 into individual PAS and GAF domains using directed mutagenesis.	[111]
iRFP670 iRFP682 iRFP702 iRFP720	643/670 nm 663/683 nm 673/702 nm 702/720 nm dimeric	Proteins with high effective brightness and low cytotoxicity in vitro and in vivo, without the addition of exogenous BV. Efficient two-color imaging in living mice. Can mainly serve for labeling of organelles and whole cells. Multicolor microscopy and whole-body imaging, photoacoustic tomography, fluorescence lifetime imaging, tumors and metastases.	Developed by mutagenesis of RpBphP6 and RpBphP2. RpBphP6 was cut to only have PAS and GAF domains, and the mutations D202 and Y258F. Finally, several rounds of random mutagenesis were performed.	[112]
miRFPs: miRFP670 miRFP703 miRFP709	642/670 nm 673/703 nm 683/709 nm monomeric	It fully relies on endogenous BV to fluoresce. miRFPs make it possible to create NIR biosensors for tasks like monitoring protein-protein interactions and other applications. They demonstrate excellent performance in microscopy, flow cytometry, and whole-body imaging.	Consists of mutated AS-GAF domains (first 315 amino acids) of RpBphP1 from *Rhodopseudomonas palustris*.	[113]
BphP1	Ex. 780 nm Em. none Max.Abs. Pfr (ON) 756 nm Pr (OFF) 678 nm monomeric	BphP1 exhibits the most significant red-shifted absorption compared to other BphPs. BphP1 has two red- and NIR-absorbing photoconvertible states, enabling photoacoustic imaging and the ability to switch between states in deep tissues.	Full-length phytochrome RpBphP1 from the bacterium *Rhodopseudomonas palustris* (~82 kDa)	[114]
mIFP	683/704 nm monomeric	mIFP effectively labels proteins in live cells without any apparent harmful effects. However, when employed in vivo, it necessitates the introduction of an external cofactor.	Modification of BrBphP from *Bradyrhizobium* with site-specific mutagenesis, DNA shuffling and random mutagenesis.	[115]
PAiRFP1 PAiRFP2	690/717 nm 692/719 nm dimeric	The weakly fluorescent PAiRFPs undergo photoconversion transforming into a strongly fluorescent state upon excitation. After photoactivation, PAiRFPs gradually return to their original state, allowing for numerous cycles of photoactivation and relaxation.	Molecular evolution of BphP from *Agrobacterium tumefaciens* C58, called AtBphP2.	[116]

**Table 3 ijms-24-14741-t003:** Examples of bacterial and fungal optoproteins used to functionalize materials.

Optoprotein	Microorganism	Functionalization Technique	Photoreceptor	Material	Wavelength	Application	Ref.
EL222	* Erythrobacter litoralis *	Thiol-malemide	EL222 Homomerization	Collagen hydrogel	450 nm Dark reversible	Enhance crossliking, improve mechanical properties for cell proliferation	[157]
CarHC	*Pectobacterium carotovorum*	Spy chemistry	CarHCc Oligomerization	Elastin-like polypeptide (ELP)	522 nm Dark reversible	Cell/protein control release and improvement of mechanical properties	[147,158]
Cph1	*Synechocystis* sp. *PCC 6803*	Michael-type addition and Histidine Tag	Cph1 Homodimerization	PEG-VS hydrogel	660 nm 740 reversible	Mechanical properties	[144,145]
VVD	*Neurospora crassa*	Histidine Tag	VVD homodimerization	Ni 2+ NTA colloidal particles	450 nm and 660 nm Dark reversible	Self-sorting /Self-assembly of colloidal particles	[145]
PYP	*Halorhodospira halophila*	Film deposition	Intramolecular conformational change	Mach–Zehnder interferometer	445 nm Dark reversible	Improve optical characteristics of IO circuits	[42]

## Data Availability

Not applicable.

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
