# Peer review of "Current Trends of Bacterial and Fungal Optoproteins for Novel Optical Applications"

_ijms, 2023, doi:10.3390/ijms241914741_

Round 1

Reviewer 1 Report

Overall, this is a well written review publication and offers a good overview summarizing the state of knowledge concerning some optoproteins from bacteria and fungi. This manuscript shows rich content, providing a deep insight for some works: the study is within the journal’s scope, and I found it to be well-written, providing sufficient information. However, before publication some points need to be clarified.

My comments:

Line 64 – please add a short methodology of this review.

Line 157 – Please check the correctness of “su Fushimi perfamily“

Line 174 – the authors should ensure that they use the term “expression” in relation to genes only

Line 191 – please change to “phrB from Agrobacterium”

Line 244 – please write ions with superscripts like “H+

Line 296 – FMN was already abbreviated in line 78

Line 366 – correct to [74].

Line 469 – “Avena sativa” should be written in italics. Also change to “Avena sativa L.”

Line 477 – the first appearance of VVD is in Table 1. Additionally, there is no need to abbreviate VVD in line 665 for the second time.

Line 520 – correct to [100].

Line 676 – please write bacteria’s names in italics.

Reviewer 2 Report

In this study, the authors presented a reviewed article entitled as, “Current Trends of Bacterial and Fungal Optoproteins for Novel Optical Applications”. The authors reviewed the researches about the photoproteins, luminescent proteins or optoproteins a kind of light-response proteins, for the conversion of light into biochemical energy that is used by some bacteria or fungi to regulate specific biological processes. That is an interesting reviewed article. However, there are some problems in that, amd the comments are as shown in the following.

1.          Please check the references. For example, in the section of Plant phytochromes (Phys), there is a label about (Pierik et al., 2021).

2.          Please shorten or rewrite the description in Table 2, especially in the column of Properties and Bacterial phytochrome template. Additionally, please check the if any plagiarism.

3.          In figure 3, the labels in the figure are capital, but the description in the legend is lower case. Please unify those.

4.          Please check the line 731, there are a repeated of [133].

The quality of English is fine. However, the author should check the sentence in the tables. 
